# TENSOR CONTRACTION & REGRESSION NETWORKS

## ABSTRACT

Convolution neural networks typically consist of many convolutional layers followed by several fully-connected layers. While convolutional layers map between high-order activation tensors, the fully-connected layers operate on flattened activation vectors. Despite its success, this approach has notable drawbacks. Flattening discards the multi-dimensional structure of the activations, and the fully-connected layers require a large number of parameters. We present two new techniques to address these problems. First, we introduce *tensor contraction layers* which can replace the ordinary fully-connected layers in a neural network. Second, we introduce *tensor regression layers*, which express the output of a neural network as a low-rank multi-linear mapping from a high-order activation tensor to the softmax layer. Both the contraction and regression weights are learned end-to-end by backpropagation. By imposing low rank on both, we use significantly fewer parameters. Experiments on the ImageNet dataset show that applied to the popular VGG and ResNet architectures, our methods significantly reduce the number of parameters in the fully connected layers (about 65% space savings) while negligibly impacting accuracy.

## 1 INTRODUCTION

Many natural datasets exhibit pronounced multi-modal structure. We represent audio spectrograms as 2nd-order tensors (matrices) with modes corresponding to frequency and time. We represent images as third-order tensors with modes corresponding to width, height and the color channels. Videos are expressed as 4th-order tensors, and the signal processed by an array of video sensors can be described as a 5th-order tensor. A broad array of multi-modal data can be naturally encoded as tensors. Tensor methods extend linear algebra to higher order tensors and are promising tools for manipulating and analyzing such data.

The mathematical properties of tensors have long been the subject of theoretical study. Previously, in machine learning, data points were typically assumed to be vectors and datasets to be matrices. Hence, spectral methods, such as matrix decompositions, have been popular in machine learning. Recently, tensor methods, which generalize these techniques to higher-order tensors, have gained prominence. One class of broadly useful techniques within tensor methods are tensor decompositions, which have been studied for learning latent variables (Anandkumar et al., 2014).

Deep Neural Networks (DNNs) frequently manipulate high-order tensors: in a standard deep convolutional Neural Network (CNN) for image recognition, the inputs and the activations of convolutional layers are $3^{\text{rd}}$-order tensors. And yet, to wit, most architectures output predictions by first flattening the activations tensors and then connecting to the output neurons via one or more fully-connected layers. This approach presents several issues: we lose multi-modal information during the flattening process and the fully-connected layers require a large number of parameters.

**In this paper**, we propose Tensor Contraction Layers (TCLs) and Tensor Regression Layers (TRLs) as end-to-end trainable components of neural networks. In doing so, we exploit multilinear structure without giving up the power and flexibility offered by modern deep learning methods. By replacing fully-connected layers with tensor contractions, we can aggregate long-range spatial information while preserving multi-modal structure. Moreover, by enforcing low rank, we can significantly reduce the number of parameters needed with minimal impact on accuracy.

Our proposed TRL represent the regression weights through the factors of a low-rank tensor decomposition. The TRL obviates the need for flattening when generating output. By combining

tensor regression with tensor contraction, we further increase efficiency. Augmenting the VGG and ResNet architectures, we demonstrate improved performance on the ImageNet dataset despite significantly reducing the number of parameters (almost by 65%). This is the first paper that presents an end-to-end trainable architecture that retains the multi-dimensional tensor structure throughout the network.

**Related work:** Several recent papers apply tensor decomposition to deep learning. Lebedev et al. (2014) propose using CP decomposition to speed up convolutional layers. Kim et al. (2015) take a pre-trained network and apply tensor (Tucker) decomposition on the convolutional kernel tensors and then fine-tune the resulting network. Yang & Hospedales (2016) propose weight sharing in multi-task learning and Chen et al. (2017) propose sharing residual units. These contributions are orthogonal to ours and can be applied together.

Novikov et al. (2015) use the Tensor-Train (TT) format to impose low-rank tensor structure on weights. However, they still retain the fully-connected layers for the output, while we present an end-to-end tensorized network architecture.

Despite the success of DNNs, many open questions remain as to why they work so well and whether they really need so many parameters. Tensor methods have emerged as promising tools of analysis to address these questions and to better understand the success of deep neural networks. Cohen et al. (2015), for example, use tensor methods as tools of analysis to study the expressive power of CNNs. Haeffele & Vidal (2015) derive sufficient conditions for global optimality and optimization of non-convex factorization problems, including tensor factorization and deep neural network training. Other papers investigate tensor methods as tools for devising neural network learning algorithms with theoretical guarantees of convergence (Sedghi & Anandkumar, 2016; Janzamin et al., 2015a;b).

Several prior papers address the power of tensor regression to preserve natural multi-modal structure and learn compact predictive models (Guo et al., 2012; Rabusseau & Kadri, 2016; Zhou et al., 2013; Yu & Liu, 2016). However, these works typically rely on analytical solutions and require manipulating large tensors containing the data. They are usually used for small dataset or require to downsampled datasets or extract compact features prior to fitting the model, and do not scale to large datasets such as ImageNet.

To our knowledge, no prior work combines tensor contraction or tensor regression with deep learning in an end-to-end trainable fashion.

## 2 MATHEMATICAL BACKGROUND

**Notation:** Throughout the paper, we define tensors as multidimensional arrays, with indexing starting at 0. First order tensors are vectors, denoted $\mathbf{v}$. Second order tensors are matrices, denoted $\mathbf{M}$ and $Id$ is the identity matrix. We denote $\tilde{\mathcal{X}}$ tensors of order 3 or greater. For a third order tensor $\tilde{\mathcal{X}}$, we denote its element $(i, j, k)$ as $\tilde{\mathcal{X}}_{i_1, i_2, i_3}$. A colon is used to denote all elements of a mode e.g. the mode-1 fibers of $\tilde{\mathcal{X}}$ are denoted as $\tilde{\mathcal{X}}_{:, i_2, i_3}$. The transpose of $\mathbf{M}$ is denoted $\mathbf{M}^\top$ and its pseudo-inverse $\mathbf{M}^\dagger$. Finally, for any $i, j \in \mathbb{N}, [i \mathinner{.\,.} j]$ denotes the set of integers $\{i, i+1, \cdots, j-1, j\}$.

**Tensor unfolding:** Given a tensor, $\tilde{\mathcal{X}} \in \mathbb{R}^{I_0 \times I_1 \times \cdots \times I_N}$, its mode-$n$ unfolding is a matrix $\mathbf{X}_{[n]} \in \mathbb{R}^{I_n, I_M}$, with $M = \prod_{\substack{k=0 \\ k \neq n}}^{N} I_k$ and is defined by the mapping from element $(i_0, i_1, \cdots, i_N)$ to $(i_n, j)$, with $j = \sum_{\substack{k=0 \\ k \neq n}}^{N} i_k \times \prod_{\substack{m=k+1 \\ m \neq n}}^{N} I_m$.

**Tensor vectorization:** Given a tensor, $\tilde{\mathcal{X}} \in \mathbb{R}^{I_0 \times I_1 \times \cdots \times I_N}$, we can flatten it into a vector $\text{vec}(\tilde{\mathcal{X}})$ of size $(I_0 \times \cdots \times I_N)$ defined by the mapping from element $(i_0, i_1, \cdots, i_N)$ of $\tilde{\mathcal{X}}$ to element $j$ of $\text{vec}(\tilde{\mathcal{X}})$, with $j = \sum_{k=0}^{N} i_k \times \prod_{m=k+1}^{N} I_m$.

**n-mode product:** For a tensor $\tilde{\mathcal{X}} \in \mathbb{R}^{I_0 \times I_1 \times \cdots \times I_N}$ and a matrix $\mathbf{M} \in \mathbb{R}^{R \times I_n}$, the n-mode product of a tensor is a tensor of size $(I_0 \times \cdots \times I_{n-1} \times R \times I_{n+1} \times \cdots \times I_N)$ and can be expressed

using unfolding of $\tilde{\mathcal{X}}$ and the classical dot product as:

$$\tilde{\mathcal{X}} \times_n \mathbf{M} = \mathbf{M}\tilde{\mathcal{X}}_{[n]} \in \mathbb{R}^{I_0 \times \cdots \times I_{n-1} \times R \times I_{n+1} \times \cdots \times I_N} \tag{1}$$

**Generalized inner-product** For two tensors $\tilde{\mathcal{X}}, \tilde{\mathcal{Y}} \in \mathbb{R}^{I_0 \times I_1 \times \cdots \times I_N}$ of same size, their inner product is defined as $\langle \tilde{\mathcal{X}}, \tilde{\mathcal{Y}} \rangle = \sum_{i_0=0}^{I_0-1} \sum_{i_1=0}^{I_1-1} \cdots \sum_{i_n=0}^{I_N-1} \tilde{\mathcal{X}}_{i_0,i_1,\cdots,i_n} \tilde{\mathcal{Y}}_{i_0,i_1,\cdots,i_n}$ For two tensors $\tilde{\mathcal{X}} \in \mathbb{R}^{D_x \times I_1 \times I_2 \times \cdots \times I_N}$ and $\tilde{\mathcal{Y}} \in \mathbb{R}^{I_1 \times I_2 \times \cdots \times I_N \times D_y}$ sharing $N$ modes of same size, we similarly defined the generalized inner product along the $N$ last (respectively first) modes of $\tilde{\mathcal{X}}$ (respectively $\tilde{\mathcal{Y}}$) as $\langle \tilde{\mathcal{X}}, \tilde{\mathcal{Y}} \rangle_N = \sum_{i_1=0}^{I_1-1} \sum_{i_2=0}^{I_1-1} \cdots \sum_{i_n=0}^{I_N-1} \tilde{\mathcal{X}}_{:,i_1,i_2,\cdots,i_n} \tilde{\mathcal{Y}}_{i_1,i_2,\cdots,i_n,:}$ with $\langle \tilde{\mathcal{X}}, \tilde{\mathcal{Y}} \rangle_N \in \mathbb{R}^{I_x,I_y}$.

**Tucker decomposition:** Given a tensor $\tilde{\mathcal{X}} \in \mathbb{R}^{I_0 \times I_1 \times \cdots \times I_N}$, we can decompose it into a low rank core $\tilde{\mathcal{G}} \in \mathbb{R}^{R_0 \times R_1 \times \cdots \times R_N}$ by projecting along each of its modes with projection factors $\left(\mathbf{U}^{(0)}, \cdots, \mathbf{U}^{(N)}\right)$, with $\mathbf{U}^{(k)} \in \mathbb{R}^{R_k, I_k}, k \in (0, \cdots, N)$.

In other words, we can write:

$$\tilde{\mathcal{X}} = \tilde{\mathcal{G}} \times_0 \mathbf{U}^{(0)} \times_1 \mathbf{U}^{(2)} \times \cdots \times_N \mathbf{U}^{(N)} = [\![\tilde{\mathcal{G}}; \mathbf{U}^{(0)}, \cdots, \mathbf{U}^{(N)}]\!] \tag{2}$$

Typically, the factors and core of the decomposition are obtained by solving a least squares problem. In particular, closed form solutions can be obtained for the factor by considering the $n-$mode unfolding of $\tilde{\mathcal{X}}$ that can be expressed as:

$$\mathbf{X}_{[n]} = \mathbf{U}^{(n)}\mathbf{G}_{[n]} \left( \mathbf{U}^{(0)} \otimes \cdots \mathbf{U}^{(n-1)} \otimes \mathbf{U}^{(n+1)} \otimes \cdots \otimes \mathbf{U}^{(N)} \right)^T \tag{3}$$

Similarly, we can optimize the core in a straightforward manner by isolating it using the equivalent rewriting of the above equality:

$$vec(\mathbf{X}) = \left( \mathbf{U}^{(0)} \otimes \cdots \otimes \mathbf{U}^{(N)} \right) vec(\mathbf{G}) \tag{4}$$

The interested reader is referred to the thorough review of the literature on tensor decompositions by Kolda & Bader (2009).

# 3 TENSOR CONTRACTION AND TENSOR REGRESSION

In this section, we explain how to incorporate tensor contractions and tensor regressions into neural networks as differentiable layers.

## 3.1 TENSOR CONTRACTION

One natural way to incorporate tensor operations into a neural network is to apply tensor contraction to an activation tensor in order to obtain a low-dimensional representation. We call this technique the Tensor Contraction layer (TCL). Compared to performing a similar rank reduction with a fully-connected layer, TCLs require fewer parameters and less computation.

**Tensor contraction layers** Given an activation tensor $\tilde{\mathcal{X}}$ of size $(S_0, D_0, D_1, \cdots, D_N)$, the TCL will produce a compact core tensor $\tilde{\mathcal{G}}$ of smaller size $(S_0, R_0, R_1, \cdots, R_N)$ defined as:

$$\tilde{\mathcal{X}}' = \tilde{\mathcal{X}} \times_0 \mathbf{V}^{(0)} \times_1 \mathbf{V}^{(1)} \times \cdots \times_N \mathbf{V}^{(N)} \tag{5}$$

with $\mathbf{V}^{(k)} \in \mathbb{R}^{R_k, I_k}, k \in [0 .. N]$. Note that the projections start at the second mode because the first mode $S_0$ corresponds to the batch.

The projection factors $\left(\mathbf{V}^{(k)}\right)_{k \in [1, \cdots N]}$ are learned end-to-end with the rest of the network by gradient backpropagation. In the rest of this paper, we denote *size*$-(R_0, \cdots, R_N)$ *TCL*, or *TCL*$-(R_0, \cdots, R_N)$ a TCL that produces a compact core of dimension $(R_0, \cdots, R_N)$.

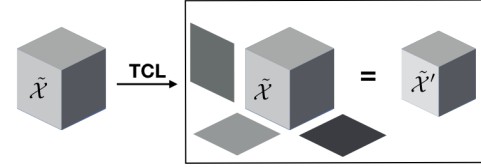

Figure 1: A representation of the Tensor Contraction Layer (TCL) on a tensor of order 3. The input tensor $\tilde{\mathcal{X}}$ is contracted into a low rank core $\tilde{\mathcal{X}}'$.

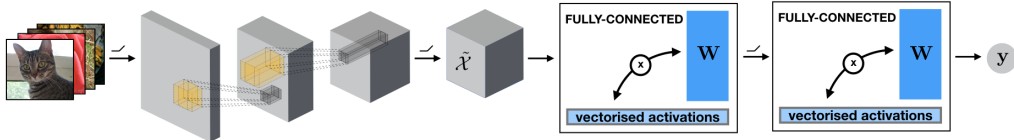

Figure 2: In standard CNNs, the input $\tilde{\mathcal{X}}$ is flattened and then passed to a fully-connected layer, where it is multiplied by a weight matrix $\mathbf{W}$.

**Gradient back-propagation**   In the case of the TCL, we simply need to take the gradients with respect to the factors $\mathbf{V}^{(k)}$ for each $k \in 0, \cdots, N$ of the tensor contraction. Specifically, we compute

$$\frac{\partial \tilde{\mathcal{X}}'}{\partial \mathbf{V}^{(k)}} = \frac{\partial \tilde{\mathcal{X}} \times_0 \mathbf{V}^{(0)} \times_1 \mathbf{V}^{(1)} \times \cdots \times_N \mathbf{V}^{(N)}}{\partial \mathbf{V}^{(k)}} \tag{6}$$

By rewriting the previous equality in terms of unfolded tensors, we get an equivalent rewriting where we have isolated the considered factor:

$$\frac{\partial \tilde{\mathcal{X}}'_{[k]}}{\partial \mathbf{V}^{(k)}} = \frac{\partial \mathbf{V}^{(k)} \mathbf{X}_{[k]} \left( Id \otimes \mathbf{V}^{(0)} \otimes \cdots \mathbf{V}^{(k-1)} \otimes \mathbf{V}^{(k+1)} \otimes \cdots \otimes \mathbf{V}^{(N)} \right)^T}{\partial \mathbf{V}^{(k)}} \tag{7}$$

**Model complexity**   Considering an activation tensor $\tilde{\mathcal{X}}$ of size $(S_0, D_0, D_1, \cdots, D_N)$, a size–$(R_0, R_1, \cdots, R_N)$ Tensor Contraction Layer will have a total of $\sum_{k=0}^{N} D_k \times R_k$ parameters.

### 3.2   LOW-RANK TENSOR REGRESSION

In order to generate outputs, CNNs typically either flatten the activations or apply a spatial pooling operation. In either case, the discard all multimodal structure, and subsequently apply a full-connected output layer. Instead, we propose leveraging the spatial structure in the activation tensor and formulate the output as lying in a low-rank subspace that jointly models the input and the output. We do this by means of a low-rank tensor regression, where we enforce a low multilinear rank of the regression weight tensor.

**Tensor regression as a layer**   Let us denote by $\tilde{\mathcal{X}} \in \mathbb{R}^{S, I_0 \times I_1 \times \cdots \times I_N}$ the input activation tensor corresponding to $S$ samples $\left( \tilde{\mathcal{X}}_1, \cdots, \tilde{\mathcal{X}}_S \right)$ and $\mathbf{Y} \in \mathbb{R}^{S,O}$ the $O$ corresponding labels for each sample. We are interested in the problem of estimating the regression weight tensor $\tilde{\mathcal{W}} \in \mathbb{R}^{I_0 \times I_1 \times \cdots \times I_N \times O}$ under some fixed low rank $(R_0, \cdots, R_N, R_{N+1})$, such that, $\mathbf{Y} = \langle \tilde{\mathcal{X}}, \tilde{\mathcal{W}} \rangle_N + b$, i.e.

$$Y = \langle \tilde{\mathcal{X}}, \tilde{\mathcal{W}} \rangle_N + \mathbf{b}$$
$$\text{subject to } \tilde{\mathcal{W}} = [\![ \tilde{\mathcal{G}}; \mathbf{U}^{(0)}, \cdots, \mathbf{U}^{(N)}, \mathbf{U}^{(N+1)} ]\!] \tag{8}$$

With $\langle \tilde{\mathcal{X}}, \tilde{\mathcal{W}} \rangle_N = \tilde{\mathcal{X}}_{[0]} \times \tilde{\mathcal{W}}_{[N+1]}$ the contraction of $\tilde{\mathcal{X}}$ by $\tilde{\mathcal{W}}$ along their $N$ last (respectively first) modes, $\tilde{\mathcal{G}} \in \mathbb{R}^{R_0 \times \cdots \times R_N \times R_{N+1}}$, $\mathbf{U}^{(k)} \in \mathbb{R}^{I_k \times R_k}$ for each $k$ in $[0 \mathinner{\ldotp\ldotp} N]$ and $\mathbf{U}^{(N+1)} \in \mathbb{R}^{O \times R_{N+1}}$.

Previously, this problem has been studied as a standalone one where the input data is directly mapped to the output, and solved analytically. However, this requires pre-processing the data to extract (hand-crafted) features to feed the model. In addition, the analytical solution is prohibitive in terms of computation and memory usage for large datasets.

In this work, we incorporate tensor regressions as trainable layers in neural networks. We do so by replacing the traditional flattening + fully-connected layers with a tensor regression applied directly to the high-order input and enforcing low rank constraints on the weights of the regression. We call our layer the *Tensor Regression Layer* (*TRL*). Intuitively, the advantage of the TRL comes from leveraging the multi-modal structure in the data and expressing the solution as lying on a low rank manifold encompassing both the data and the associated outputs.

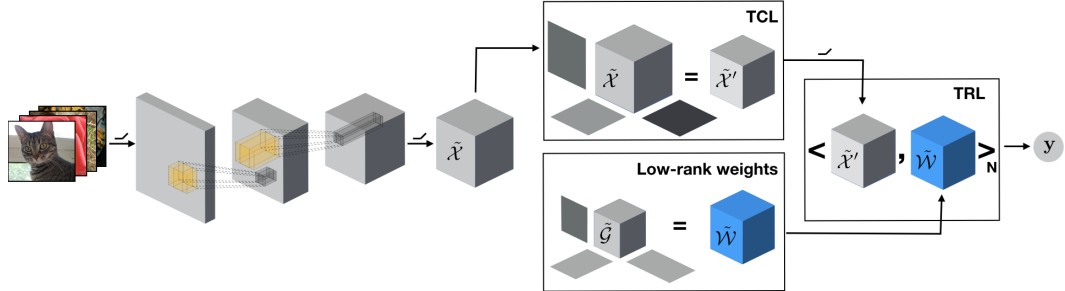

Figure 3: We propose to first reduce the dimensionality of the activation tensor by applying tensor contraction before performing tensor regression. We then replace flattening operators and fully-connected layers by a TRL. The output is a product between the activation tensor and a low-rank weight tensor $\tilde{\mathcal{W}}$. For clarity, we illustrate the case of a binary classification, where $y$ is a scalar. For multi-class, $y$ becomes a vector and the regression weights would become a $4^{th}$ order tensor.

**Gradient backpropagation**   The gradients of the regression weights and the core with respect to each factor can be obtained by writing:

$$\frac{\partial \tilde{\mathcal{W}}}{\partial \mathbf{U}^{(k)}} = \frac{\partial \tilde{\mathcal{G}} \times_0 \mathbf{U}^{(0)} \times_1 \mathbf{U}^{(1)} \times \cdots \times_{N+1} \mathbf{U}^{(N+1)}}{\partial \mathbf{U}^{(k)}} \tag{9}$$

Using the unfolded expression of the regression weights, we obtain the equivalent formulation:

$$\frac{\partial \tilde{\mathcal{W}}_{[k]}}{\partial \mathbf{U}^{(k)}} = \frac{\partial \mathbf{U}^{(k)} \mathbf{G}_{[k]} \left( \mathbf{U}^{(0)} \otimes \cdots \mathbf{U}^{(k-1)} \otimes \mathbf{U}^{(k+1)} \otimes \cdots \otimes \mathbf{U}^{(N+1)} \right)^T}{\partial \mathbf{U}^{(k)}} \tag{10}$$

Similarly, we can obtain the gradient with respect to the core by considering the vectorized expressions:

$$\frac{\partial \mathrm{vec}(\tilde{\mathcal{W}})}{\partial \mathrm{vec}(\tilde{\mathcal{G}})} = \frac{\partial \left( \mathbf{U}^{(0)} \otimes \cdots \otimes \mathbf{U}^{(N+1)} \right) vec(\mathbf{G})}{\partial \mathrm{vec}(\tilde{\mathcal{G}})} \tag{11}$$

**Model analysis**   We consider as input an activation tensor $\tilde{\mathcal{X}} \in \mathbb{R}^{S, I_0 \times I_1 \times \cdots \times I_N}$, and a rank-$(R_0, R_1, \cdots, R_N, R_{N+1})$ tensor regression layer, where, typically, $R_k \leq I_k$. Let's assume the output is $n$-dimensional. A fully-connected layer taking $\tilde{\mathcal{X}}$ as input will have $n_{\mathrm{FC}} = n \times \prod_{k=0}^{N} I_k$ parameters.

By comparison, the TRL has a number of parameters $n_{\mathrm{TRL}}$, with:

$$n_{\mathrm{TRL}} = \prod_{k=0}^{N+1} R_k + \sum_{k=0}^{N} R_k \times I_k + R_{N+1} \times n \tag{12}$$

## 4   EXPERIMENTS

We empirically demonstrate the effectiveness of preserving the tensor structure through tensor contraction and tensor regression by integrating it into state-of-the-art architectures and demonstrating similar performance on the popular ImageNet dataset. In particular, we empirically verify the effectiveness of the TCL on VGG-19 (Simonyan & Zisserman, 2014) and conduct thorough experiment with the tensor regression on ResNet-50 and ResNet-101 (He et al., 2015).

### 4.1   EXPERIMENTAL SETTING

**Synthetic data**   To illustrate the effectiveness of the low-rank tensor regression, we first apply it to synthetic data $y = \mathrm{vec}(\tilde{\mathcal{X}}) \times \mathbf{W}$ where each sample $\tilde{\mathcal{X}} \in \mathbb{R}^{(64)}$ follows a Gaussian distribution

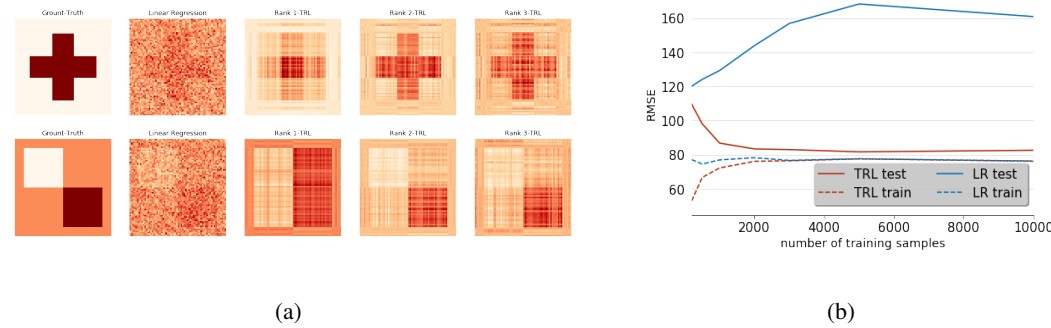

(a)                                                                                       (b)

Figure 4: Empirical comparison (4a) of the TRL against regression with a fully-connected layer. We plot the weight matrix of both the TRL and a fully-connected layer. Due to its low-rank weights, the TRL better captures the structure in the weights and is more robust to noise. Evolution of the RMSE as a function of the training set size (4b) for both the TRL and fully-connected regression

Table 1: Results obtained on ImageNet by adding a TCL to a VGG-19 architecture. We reduce the number of hidden units proportionally to the reduction in size of the activation tensor following the tensor contraction. Doing so allows more than 65% space savings over all three fully-connected layers (i.e. 99.8% space saving over the fully-connected layer replaced by the TCL) with no corresponding decrease in performance (comparing to the standard VGG network as a baseline).

| Method | | Accuracy | | Space Savings |
|---|---|---|---|---|
| TCL–size | Hidden Units | Top-1 (%) | Top-5 (%) | (%) |
| baseline | 4096 | 68.7 | 88 | 0 |
| (512, 7, 7) | 4096 | **69.4** | **88.3** | -0.21 |
| (384, 5, 5) | 3072 | 68.3 | 87.8 | **65.87** |

$\mathcal{N}(0, 3)$. $\mathbf{W}$ is a fixed matrix and the labels are generated as $y = \text{vec}(\tilde{\mathcal{X}}) \times \mathbf{W}$. We then train the data on $\tilde{\mathcal{X}} + \tilde{\mathcal{E}}$, where $\tilde{\mathcal{E}}$ is added Gaussian noise sampled from $\mathcal{N}(0, 3)$. We compare i) a TRL with squared loss and ii) a fully-connected layer with a squared loss. In Figure 4a, we show the trained weight of both a linear regression based on a fully-connected layer and a TRL with various ranks, both obtained in the same setting. As can be observed in Figure 5b, the TRL is easier to train on small datasets and less prone to over-fitting, due to the low rank structure of its regression weights, as opposed to typical Fully Connected based Linear Regression.

**ImageNet Dataset**    We ran our experiments on the widely-used ImageNet-1K dataset, using several widely-popular network architectures. The ILSVRC dataset (ImageNet) is composed of 1.2 million images for training and 50, 000 for validation, all labeled for 1,000 classes. Following (Huang et al., 2016a; He et al., 2015; Huang et al., 2016b; He et al., 2016), we report results on the validation set in terms of Top-1 accuracy and Top-5 accuracy across all 1000 classes. Specifically, we evaluate the classification error on single $224 \times 224$ single center crop from the raw input images.

**Training the TCL + TRL**    When experimenting with the tensor regression layer, we did not retrain the whole network each time but started from a pre-trained ResNet. We experimented with two settings: i) We replaced the last average pooling, flattening and fully-connected layer by either a TRL or a combination of TCL + TRL and trained these from scratch while keeping the rest of the network fixed. ii) We investigate replacing the pooling and fully-connected layers with a TRL that jointly learns the spatial pooling as part of the tensor regression. In that setting, we also explore initializing the TRL by performing a Tucker decomposition on the weights of the fully-connected layer.

**Implementation details**    We implemented all models using the MXNet library (Chen et al., 2015) and ran all experiments training with data parallelism across multiple GPUs on Amazon Web Services, with 4 NVIDIA k80 GPUs. For training, we adopt the same data augmentation procedure as in the original Residual Networks (ResNets) paper (He et al., 2015).

When training the layers from scratch, we found it useful to add a batch normalization layer (Ioffe & Szegedy, 2015) before and after the TCL/TRL to avoid vanishing or exploding gradients, and to make the layers more robust to changes in the initialization of the factors. In addition we constrain the weights of the tensor regression by applying $\ell_2$ normalization (Salimans & Kingma, 2016) to the factors of the Tucker decomposition.

## 4.2    RESULTS

Table 2: Results obtained with ResNet-50 on ImageNet. The first row corresponds to the standard ResNet. Rows 2 and 3 present the results obtained by replacing the last average pooling, flattening and fully-connected layers with a TRL. In the last row, we have also added a TCL.

| | Method | | Accuracy | |
| --- | --- | --- | --- | --- |
| Architecture | TCL–size | TRL rank | Top-1 (%) | Top-5 (%) |
| Resnet-50 | baseline with spatial pooling | | 74.58 | 92.06 |
| | - | (1000, 2048, 7, 7) | 73.6 | 91.3 |
| | - | (500, 1024, 3, 3) | 72.16 | 90.44 |
| | (1024, 3, 3) | (1000, 1024, 3, 3) | 73.43 | 91.3 |
| Resnet-101 | baseline with spatial pooling | | 77.1 | 93.4 |
| | - | (1000, 2048, 7, 7) | 76.45 | 92.9 |
| | - | (500, 1024, 3, 3) | 76.7 | 92.9 |
| | (1024, 3, 3) | (1000, 1024, 3, 3) | 76.56 | 93 |

**Impact of the tensor contraction layer**    We first investigate the effectiveness of the TCL using a VGG-19 network architecture (Simonyan & Zisserman, 2014). This network is especially well-suited for out methods because of its $138, 357, 544$ parameters, $119, 545, 856$ of which (more than 80% of the total number of parameters) are contained in the fully-connected layers. By adding TCL to contract the activation tensor prior to the fully-connected layers we can achieve large space saving. We can express the space saving of a model $M$ with $n_M$ total parameters in its fully-connected layers with respect to a reference model $R$ with $n_R$ total parameters in its fully-connected layers as $1 - \frac{n_M}{n_R}$ (bias excluded).

Table 1 presents the accuracy obtained by the different combinations of TCL in terms of top-1 and top-5 accuracy as well as space saving. By adding a TCL that preserves the size of its input we are able to obtain slightly higher performance with little impact on the space saving (0.21% of space loss) while by decreasing the size of the TCL we got more than $65\%$ space saving with almost no performance deterioration.

**Overcomplete TRL**    We first tested the TRL with a ResNet-50 and a ResNet-101 architectures on ImageNet, removing the average pooling layer to preserve the spatial information in the tensor. The full activation tensor is directly passed on to a TRL which produces the outputs on which we apply softmax to get the final predictions. This results in more parameters as the spatial dimensions are preserved. To reduce the computational burden but preserve the multi-dimensional information, we alternatively insert a TCL before the TRL. In Table 2, we present results obtained in this setting on ImageNet for various configurations of the network architecture. In each case, we report the size of the TCL (i.e. the dimension of the contracted tensor) and the rank of the TRL (i.e. the dimension of the core of the regression weights).

**Joint spatial pooling and low-rank regression**    Alternatively, we can learn the spatial pooling as part of the tensor regression. In this case, we remove the average pooling layer and feed the tensor

Table 3: Results obtained with a ResNet-101 architecture on ImageNet, learning spatial pooling as part of the TRL.

| TRL rank | Performance (%) | | |
|---|---|---|---|
| | Top-1 | Top-5 | Space savings |
| baseline | **77.1** | **93.4** | 0 |
| (200, 1, 1, 200) | **77.1** | **93.2** | 68.2 |
| (150, 1, 1, 150) | 76 | 92.9 | 76.6 |
| (100, 1, 1, 100) | 74.6 | 91.7 | 84.6 |
| (50 , 1, 1, 50) | 73.6 | 91 | **92.4** |

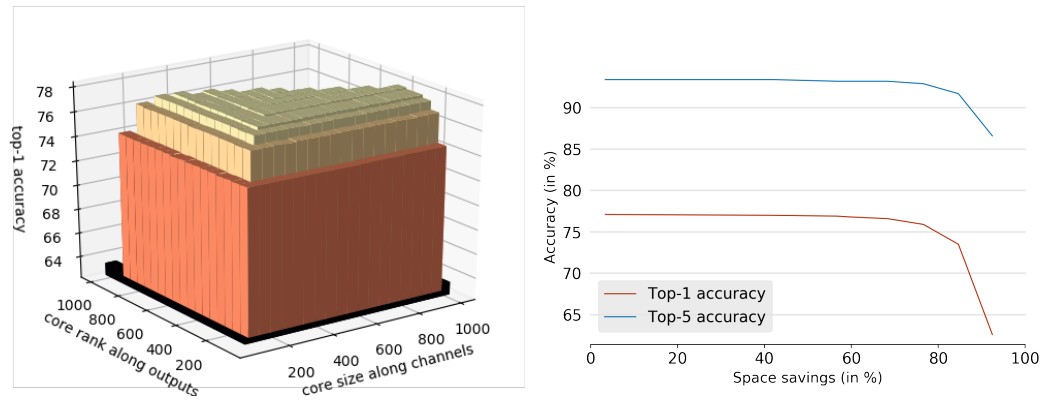

(a) Accuracy as a function of the core size          (b) Accuracy as a function of space savings

Figure 5: 5a shows the Top-1 accuracy (in %) as we vary the size of the core along the number of outputs and number of channels (the TRL does spatial pooling along the spatial dimensions, i.e., the core has rank 1 along these dimensions).

of size (batch size, number of channels, height, width) to the TRL, while imposing a rank of 1 on the spatial dimensions of the core tensor of the regression. Effectively, this setting simultaneously learns weights for the multi-linear spatial pooling as well as the regression.

In practice, to initialize the weights of the TRL in this setting, we consider the weight of fully-connected layer from a pre-trained model as a tensor of size (batch size, number of channels, 1, 1, number of classes) and apply a partial tucker decomposition to it by keeping the first dimension (batch-size) untouched. The core and factors of the decomposition then give us the initialization of the TRL. The projection vectors over the spatial dimension are then initialize to $\frac{1}{height}$ and $\frac{1}{width}$, respectively. The Tucker decomposition was performed using TensorLy (Kossaifi et al., 2016). In this setting, we show that we can drastically decrease the number of parameters with little impact on performance. In Figure 5, we show the change of the Top-1 and Top-5 accuracy as we decrease the size of the core tensor of the TRL and also the space savings.

## 5 CONCLUSIONS

Unlike fully-connected layers, TCLs and TRLs obviate the need to flatten input tensors. Our experiments demonstrate that by imposing a low-rank constraint on the weights of the regression, we can learn a low-rank manifold on which both the data and the labels lie. The result is a compact network, that achieves similar accuracies with many fewer parameters. Going forward, we plan to apply the TCL and TRL to more network architectures. We also plan to leverage recent work (Shi et al., 2016) on extending BLAS primitives to avoid transpositions needed when computing tensor contractions.

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
