# OpenReview forum: "Tensor Contraction & Regression Networks"
_ICLR.cc/2018/Conference — Reject_

### Official Review · AnonReviewer2 · 2017-11-24
**Interesting but contributions are not enough**

**Rating:** 4
**Confidence:** 4

**Review:**

In this paper, new layer architectures of neural networks using a low-rank representation of tensors are proposed. The main idea is assuming Tucker-type low-rank assumption for both a weight and an input. The performance is evaluated with toy data and Imagenet.

[Clarity]
The paper is well written and easy to follow.

[Originality]
I mainly concern about the originality. Applying low-rank tensor decomposition in a network architecture has a lot of past studies and I feel this paper fails to clarify what is really distinguished from the other studies. For example, I found at least two papers [1,2] that are relevant. ([2] appears at the reference but it is not referred to.) How is the proposed method different from them?

Also, the "end-to-end" feature is repeatedly emphasized in the paper, but I don't understand its benefit.

[1] Tai, Cheng, et al. "Convolutional neural networks with low-rank regularization." arXiv preprint arXiv:1511.06067 (2015).
[2] Lebedev, Vadim, et al. "Speeding-up convolutional neural networks using fine-tuned cp-decomposition." arXiv preprint arXiv:1412.6553 (2014).

[Significance]
In the experiments, the proposed method is compared with the vanilla model (i.e., the model having no low-rank structure) but with no other baseline using different compression techniques such as Novikov et al., 2015. So I cannot judge whether this method is better in terms of compression-accuracy tradeoff.


Pros:
- The proposed model (layer architecture) is simple and easy to implement

Cons:
- The novelty is low
- No competitive baseline in experiments

---

> ### Author Response · Authors · 2018-01-04
> **Response to AnonReviewer2**
>
> We thank the reviewer for the feedback and offer some clarifications regarding the primary criticisms:
>
> The two publications that you mention focus on re-parametrizing convolutional layers, with the main purpose of speeding these up:
> [1] (Convolutional neural networks with low-rank regularization) parametrizes each convolutional layers as the composition of two convolutional layers with less parameters.
> [2] (Speeding-up convolutional neural networks using fine-tuned cp-decomposition) is referred to in the related work and also focuses on speeding up convolutional layers. This is done by performing CP decomposition on the convolution kernel before fine-tuning the whole network.
>
> These papers only focus on decomposing the weight tensors. By contrast, we propose to preserve the multi-linear structure of the activation throughout the network. In particular TCL focuses on contraction activation tensors. Additionally, those works focus on the convolutional kernels, where the number of parameters is already quite small, while our work is focused on eliminating the standard flattening and fully-connected layers
>
> Note that in the TCL, we do not assume a Tucker form of the activation tensors but rather apply tensor contraction to them to reduce their dimensionality. Similarly, TRL is a new layer that does not simply consist in assuming a Tucker form of the regression weight but directly maps an input tensor to an output tensor using low-rank regression tensor weights. It can be used to replace the flattening and fully-connected layers in traditional network architectures.
>
> Our contribution is the introduction of these two novel layers, trainable end-to-end using gradient backpropagation. Being able to these train end-to-end is crucial to be able to learn the whole network jointly (most existing tensor methods are solved analytically, and existing work on deep learning and tensor decomposition focuses mainly on pre-training a network, applying some sort of decomposition to the weights and fine-tuning. By training end-to-end we learn the whole network jointly).
>
> We compare with a competitive, state-of-the-art architecture, (ResNet) and show that performance is maintained while showing large space savings. Note that there has been no such attempt in the past to compare with. As we mention in the related work, Novikov et. al. (2015) retain the flattening and fully connected layers for the output while we present an end-to-end tensorized architecture. They obtain space savings by applying tensor decomposition to the weights of some of the fully-connected layers, reshaped as tensors (which also means selecting both the size of the tensor to reshape to, in addition to the rank of the decomposition).

---

### Official Review · AnonReviewer3 · 2017-11-25
**see detailed review below**

**Rating:** 6
**Confidence:** 4

**Review:**

This paper incorporates tensor decomposition and tensor regression into CNN by replacing its flattening operations and fully-connected layers with a new tensor regression layer.

Pros:

The low-rank representation of tensors is able to reduce the model complexity in the original CNN without sacrificing much prediction accuracy. This is promising as it enables the implementation of complex deep learning algorithms on mobile devices due to its huge space saving performance.  Overall, this paper is easy to follow.

Cons:

Q1: Can the authors discuss the computational time of the proposed tensor regression layers and compare it to that of the baseline CNN? The tensor regression layer is computationally more expensive than the flattening operations in original CNN. Usually, it also involves expensive model selection procedure to choose the tuning parameters (N+1 ranks and a L2 norm sparsity parameter). In the experiments, the authors simply tried a few ranks without serious tuning.

Q2: The authors reported the space saving in Table 1 but not in Table 2. Since spacing saving is a major contribution of the proposed method, can authors add the space saving percentage in Table 2?

Q3: There are a few typos in the current paper. I would suggest the authors to take a careful proofreading. For example,

(1) In the “Related work“ paragraph on page 2, “Lebedev et al. (2014) proposes…” should be “Lebedev et al. (2014) propose…”. Many other references have the same issue.

(2) In Figure 1, the letter $X$ should be $\tilde{\cal X}$.

(3) In expression (5) on page 3, the core tensor is denoted by $\tilde{\cal G}$. Is this the same as $\tilde{\cal X}^{‘}$ in Figure 1?

(4) In expression (5) on page 3, the core tensor $\tilde{\cal G}$ is of dimension $(D_0, R_1, \ldots, R_N)$. However, in expression (8) on page 5, $\tilde{\cal G}$ is of dimension $(R_0, R_1, \ldots, R_N, R_{N+1})$.

(5) Use \cite{} and \citep{} correctly. For example, in the “Related work“ paragraph on page 2,

“Several prior papers address the power of tensor regression to preserve natural multi-modal structure and learn compact predictive models Guo et al. (2012); Rabusseau & Kadri (2016); Zhou et al. (2013); Yu & Liu (2016).”

should be

“Several prior papers address the power of tensor regression to preserve natural multi-modal structure and learn compact predictive models (Guo et al., 2012; Rabusseau & Kadri, 2016; Zhou et al., 2013; Yu & Liu, 2016).”

---

> ### Author Response · Authors · 2018-01-04
> **Response to AnonReviewer3**
>
> We thank the reviewer for the feedback and address each point below:
>
> Q1: We show (Figure 5) that there is a large region where the rank can be decreased without impacting performance, making rank selection easy. In particular, we plot the evolution of the performance as a function of the rank. Please note that l2 normalization, which does not add extra parameters to tune (the parameters regularization is done via weight decay, as done in all state-of-the-art architectures, we kept the same parameters as in the original architectures).
>
> Q2: Table 2 corresponds to the overcomplete case (without pooling), it therefore didn’t make as much sense to mention the space savings since the corresponding architecture is not a standard used one. The main point of that experiment is to show that tensor contraction and regression can be used not only for low-rank problems but also over-defined ones (i.e. by leveraging the low-rank structure of the tensors we can optimise efficiently larger networks).
>
> Q3: Thank you for pointing these out, they have all been corrected:
>    (1), (2) and (5) have been corrected
>    (3): yes, they referred to the same, we have changed the notation to clarify this.
>    (4): this notation has also been clarified. The idea is that we leave the first dimension (batch size) untouched in both the TCL and TRL. In the TCL, X’ corresponds to the output of the layer while in the TRL, it corresponds to the core of the regression weights and therefore does not include the batch size. We now denote X’ the output of the TCL, while G denotes the core of the tensor regression weights.

---

### Official Review · AnonReviewer1 · 2017-11-25
**Contribution seems less.**

**Rating:** 4
**Confidence:** 3

**Review:**

This paper combines the tensor contraction method and the tensor regression method and applies them to CNN. This paper is well written and easy to read.

However, I cannot find a strong or unique contribution from this paper. Both of the methods (tensor contraction and tensor decomposition) are well developed in the existing studies, and combining these ideas does not seem non-trivial.

--Main question

Why authors focus on the combination of the methods? Both of the two methods can perform independently. Is there a special synergy effect?

--Minor question

The performance of the tensor contraction method depends on a size of tensors. Is there any effective way to determine the size of tensors?

---

> ### Author Response · Authors · 2018-01-04
> **Response to AnonReviewer1**
>
> We thank the reviewer for the feedback.
>
> 1. Regarding the novelty of this work: To our knowledge, no previous papers propose incorporating either tensor contraction or regression as layers in deep neural networks. Instead these methods have previously been studied as stand-alone techniques, where they are solved analytically. Our main contribution is the introduction of these two novel layers, trainable end-to-end via gradient descent and the empirical finding that we can enjoy dramatic space savings with negligible loss in accuracy.
>
> It is possible that the reviewer has encountered pre-printed versions of this paper. To preserve double-blindness, we won’t link to those drafts here. But we request that the reviewer be careful not to mistakenly hold against this work its previous inclusion in workshops and on the arXiv.
>
> 2. Regarding the usefulness of studying the two in combination: We combine the two as they are naturally complementary methods. Tensor contraction reduces the dimensionality of the input tensor, this reduced tensor can then be mapped to an output tensor using tensor regression.
>
> As shown in figure 5, the performance of the TRL is not very sensitive to the choice of the rank, making that selection easy.

---

### Public Comment · (anonymous) · 2017-11-28
**Related Work**

In what way is the proposed tensor contraction layer different from Tucker Decomposition for Feature Fusion in 'Attribute-Enhanced Face Recognition with Neural Tensor Fusion Networks'[1]? How does the tensor regression layer compare with the nonlinear activations in 'Factorized Bilinear Models for Image Recognition'[2] and 'Non-linear Convolution Filters for CNN-based Learning'[3]?

[1] http://www.research.ed.ac.uk/portal/en/publications/attributeenhanced-face-recognition-with-neural-tensor-fusion-networks(b5f001b1-21c5-44e0-ad6f-21481e83590e).html

[2] https://arxiv.org/abs/1611.05709

[3] https://arxiv.org/abs/1708.07038

---

> ### Author Response · Authors · 2018-01-04
> **Response to anonymous comment**
>
> Thank you for your interest in our paper.
>
> Regarding the methods you mention:
>
> [3] (Non-linear Convolution Filters for CNN-based learning) is a method to augment existing architectures by exploring a combination of linear and non-linear filters in the convolutional layers.
>
> [2] (Factorized Bilinear Models for Image Recognition) is in the well studied field of bilinear models. Tensor Contraction can be seen as a generalisation of bilinear pooling to any arbitrary number of dimensions.
>
> [1] (Attribute-Enhanced Face Recognition with Neural Tensor Fusion) proposes a feature fusion method as a tensor optimisation problem. Specifically, it performs fusion from two feature vectors and the framework is therefore limited to a third order weight tensor with two vector inputs.
>
> Our work is significantly different from these:
> We propose to preserve and leverage the tensor structure of the activations. We do so by introducing new generic, end-to-end trainable layers that allow large space savings while preserving the multi-dimensional structure. Specifically, we introduce Tensor Contraction Layers (TCL) that reduce the dimension of the input while preserving its multi-linear structure, and Tensor Regression Layer (TRL) that directly maps an input tensor to an output tensor using low-rank regression weights.

---

### Decision · Program_Chairs · 2018-01-29
**ICLR 2018 Conference Acceptance Decision**

**Decision:**

Reject

**Comment:**

This paper proposes methods for replacing parts of neural networks with tensors, the values of which are efficiently estimated through factorisation methods. The paper is well written and clear, but the two main objections from reviewers surround the novelty and evaluation of the method proposed. I am conscious that the authors have responded to reviewers on the topic of novelty, but the case could be made more strongly in the paper, perhaps by showing significant improvements over alternatives. The evaluation was considered weak by reviewers, in particular due to the lack of comparable baselines.

Interesting work, but I'm afraid on the basis of the reviews, I must recommend rejection.